# New Neonatal and Prenatal Approach to Home Therapy with Amoxicillin, Rifaximin, and Anti-Inflammatory Drugs for Pregnant Women with COVID-19 Infections—Monitoring of Fetal Growth as a Prognostic Factor: A Triple Case Series (N.A.T.H.A.N.)

**DOI:** 10.3390/biomedicines13081858

**Published:** 2025-07-30

**Authors:** Carlo Brogna, Grazia Castellucci, Elrashdy M. Redwan, Alberto Rubio-Casillas, Luigi Montano, Gianluca Ciammetti, Marino Giuliano, Valentina Viduto, Mark Fabrowski, Gennaro Lettieri, Carmela Marinaro, Marina Piscopo

**Affiliations:** 1Craniomed Group Srl, Microbiology Research Facility, 83038 Montemiletto, Italy; 2Castellucci OralMed Srl, Medical Clinic, 86170 Isernia, Italy; 3Department of Biological Sciences, Faculty of Science, King Abdulaziz University, Jeddah 21589, Saudi Arabia; lradwan@kau.edu.sa; 4Therapeutic and Protective Proteins Laboratory, Protein Research Department, Genetic Engineering and Biotechnology Research Institute, City for Scientific Research and Technology Applications, New Borg El-Arab 21934, Egypt; 5Autlan Regional Hospital, Jalisco Health Services, Autlan 48900, Jalisco, Mexico; alberto.rubio@sems.udg.mx; 6Andrology Unit and Service of Lifestyle Medicine in Uro-Andrology, Local Health Authority (ASL), 84124 Salerno, Italy; luigimontano@gmail.com; 7Otorhinolaryngology Unit, Hospital Ferdinando Veneziale Isernia, Regional Health Authority of Molise, 86170 Isernia, Italy; gciammetti@virgilio.it; 8Maraconsulting Srl, Public Health Company, 80128 Naples, Italy; marino@marsanconsulting.it; 9Long COVID-19 Foundation, Leeds LS25 1NB, UK; v.viduto@longcovidcharity.org; 10Department of Emergency Medicine, Royal Sussex County Hospital, University Hospitals Sussex, Brighton BN2 5BE, UK; 11Department of Biology, University of Naples Federico II, 80126 Naples, Italy; gennarole@outlook.com (G.L.); carmelamarinaro9@gmail.com (C.M.)

**Keywords:** ultrasound, pregnancy, viral infection, microbiome, fetal growth, COVID-19, pandemic

## Abstract

**Background:** Since the COVID-19 pandemic, managing acute infections in symptomatic individuals, regardless of vaccination status, has been widely debated and extensively studied. Even more concerning, however, is the impact of COVID-19 on pregnant women—especially its effects on fetuses and newborns. Several studies have documented complications in both expectant mothers and their infants following infection. **Methods:** In our previous works, we provided scientific evidence of the bacteriophage behavior of SARS-CoV-2 (Severe Acute Respiratory Syndrome Coronavirus 2). This demonstrated that a well-defined combination of two antibiotics, amoxicillin and rifaximin, is associated with the same statistics for subjects affected by severe cases of SARS-CoV-2, regardless of vaccination status. We considered the few cases in the literature regarding the management of pregnancies infected with SARS-CoV-2, as well as previous data published in our works. In this brief case series, we present two pregnancies from the same unvaccinated mother—one prior to the COVID-19 pandemic and the other during the spread of the Omicron variant—as well as one pregnancy from a mother vaccinated against COVID-19. We describe the management of acute maternal infection using a previously published protocol that addresses the bacteriophage and toxicological mechanisms associated with SARS-CoV-2. **Results:** The three pregnancies are compared based on fetal growth and ultrasound findings. This report highlights that, even in unvaccinated mothers, timely and well-guided management of symptomatic COVID-19 can result in positive outcomes. In all cases, intrauterine growth remained within excellent percentiles, and the births resulted in optimal APGAR scores. **Conclusions:** This demonstrates that a careful and strategic approach, guided by ultrasound controls, can support healthy pregnancies during SARS-CoV-2 infection, regardless of vaccination status.

## 1. Introduction

During the COVID-19 pandemic, a wide range of treatment strategies were used, particularly in the early stages, when hospital intensive care units (ICUs) were under immense strain. The international community, led by the World Health Organization (WHO), issued guidelines that shaped public health policies across both European and non-European countries. Italy, as the first nation outside China to face a major outbreak, became a critical reference point for the global response. Now, five years later, no clear consensus has emerged on the optimal treatment strategy. Scientific debates and divergent opinions continue, despite numerous studies investigating the virus’s pathophysiological mechanisms and potential therapies. While mass vaccination—primarily aimed at eliciting antibodies against the viral spike protein [1,2,3,4,5]—was widely promoted as the central solution, other research efforts have focused on novel antiviral drugs designed to inhibit viral replication within host cells [6,7,8,9,10,11]. Additionally, emerging evidence has highlighted the significant role of the human microbiome in SARS-CoV-2 infection. Studies have demonstrated that the virus can profoundly alter the gut microbiome, leading to dysbiosis, an imbalance in microbial populations that can weaken immune responses and exacerbate disease severity [12,13,14,15,16,17,18,19,20,21,22]. These findings suggest that addressing microbiome health could be a critical element in the management of COVID-19 and Long COVID Syndrome and in improving patient outcomes. Through more than 10 of our studies, we have bridged a historical gap and demonstrated that, for certain RNA viruses such as SARS-CoV-2 and Poliovirus (PLV) [23,24], the bacteriophage event represents the first true pathogenic moment, both logically and spatially. Our studies have involved molecular genetics, electron microscopy, immunofluorescence microscopy, and particularly the use of ^15^N radioisotopes incorporated into bacterial cultures derived from patient microbiota. Several research studies have shown that SARS-CoV-2, or other viruses like HIV (Human Immunodeficiency Virus), can infect gut bacteria and exhibit bacteriophage-like behavior [25,26,27,28,29,30,31,32]. Further studies have shown that the virus can persist and replicate independently for over 30 days in cultured bacteria obtained from patient feces [31]. This discovery led to the investigation of antibiotic therapy and, in particular, some antibiotics (amoxicillin and rifaximin) tested with appropriate antibiograms as a potential treatment for COVID-19 infection [25,26,27,28,29,30,31,32]. Early administration of amoxicillin and rifaximin was found to significantly reduce recovery time and support sustained blood oxygen saturation levels, whereas delayed initiation of antibiotics—particularly, after the first three days—was associated with an increased risk of pneumonia in both vaccinated and unvaccinated patients. Despite these ground-breaking findings, the role of viral replication in prokaryotic cells remains largely unrecognized by the mainstream scientific community. It must be noted, unfortunately, that proper scientific rigor should have mandated these controls from the outset, rather than defining the pathogen as eukaryotic without first excluding bacteria—the most dominant life forms on the planet. Overlooking this aspect may impact the success of treatment strategies and influence critical decisions regarding vaccination approaches and target selection. One of the most challenging aspects of managing COVID-19 cases in hospitals, universities, and clinical settings has been the management of acute SARS-CoV-2 infection during pregnancy. Regardless of whether the expectant mother has vaccine-induced or natural immunity [32], the acute phase of COVID-19 requires careful monitoring to mitigate the risks to both the mother and the unborn child. Given the heightened sensitivity of this topic, antivirals [33] have been recommended for the general population in moderate-to-severe cases if administered within five days of symptom onset. However, their use during pregnancy has been limited due to potential risks, prompting some researchers to investigate monoclonal antibody therapy as an alternative [34]. Managing pregnancy in the context of COVID-19 is especially complex, particularly given that SARS-CoV-2 exhibits bacteriophage-like properties [30]. A deeper understanding of this unique viral behavior may offer valuable insights for developing improved treatment approaches and pregnancy care strategies during the ongoing pandemic. Numerous studies have assessed the efficacy of SARS-CoV-2 vaccines administered during pregnancy, focusing primarily on infection rates among vaccinated mothers [35]. A large meta-analysis involving 30 studies and a total of 862,272 pregnant women found that vaccination was associated with a 60% reduction in infection rates, a 53% decrease in COVID-19-related hospitalizations, and an 82% reduction in intensive care unit (ICU) admissions. However, the analysis also revealed that infants born to vaccinated mothers during the Omicron wave had a 1.78-fold increased risk of infection within their first six months [35]. Additionally, the meta-analysis indirectly indicates that despite vaccination, 40% of mothers contracted the virus, 47% required hospitalization, and 18% were admitted to the ICU [35]. This suggests that although vaccination provided some protection, it did not completely prevent severe outcomes in all cases. It is important to note that the vaccination strategies analyzed in these studies focused primarily on increasing antibody levels to inhibit the binding of the SARS-CoV-2 spike protein to epithelial cells. These approaches have largely overlooked emerging research on alternative viral pathways, particularly the interaction between SARS-CoV-2 and the microbiota, which may play a pivotal role in disease progression and the immune response [27]. Understanding these mechanisms may provide new perspectives for optimizing maternal and neonatal protection against COVID-19.

COVID-19 infection during pregnancy presents a complex and unpredictable challenge, as disease progression and outcomes can vary significantly. Early studies conducted during the pandemic, including one of the first case series from Wuhan, examined the effects of COVID-19 in pregnant women during their third trimester. The study reported several complications among newborns: 17% were preterm births, 6% had mild neonatal asphyxia, 6% had neonatal gastrointestinal hemorrhage, 6% had necrotizing enteritis, 11% had hyperbilirubinemia, and 6% had neonatal diarrhea. Meanwhile, 28% had bacterial pneumonia [36]. In the study, mothers were hospitalized and treated with antivirals, and in cases of bacterial infection, with antibiotics [36]. The scientific literature continues to raise significant concerns regarding COVID-19 infection during pregnancy. Early findings from 2020 revealed histopathological evidence of vascular distress in postnatal placental examinations [37,38]. Data from another meta-analysis show that premature birth occurred in 29.7% and 16% of case series and observational studies, respectively. There were a total of three stillbirths and two neonatal deaths [39]. These findings emphasize the need for close monitoring and comprehensive management strategies in pregnant women with COVID-19 to minimize risks to both mother and child.

A Portuguese study examining 12 pregnant women infected with SARS-CoV-2 confirmed, as many others have, that most patients were generally asymptomatic. However, the study also reported the following significant fetal outcomes: one stillbirth, one case of preterm prelabor rupture of membranes, and one case of fetal growth restriction [40]. Similarly, during the first wave of COVID-19 in Italy, data collected by the Superior Institute of Health from 548 pregnant women provided further insight into maternal and neonatal outcomes. Among the 428 mothers who gave birth, four (1%) newborns were stillborn, 2.3% of newborns had severe neonatal morbidity, 14.7% were admitted to the Neonatal Intensive Care Unit (NICU), and 13.6% had a birth weight of less than 2500 g. Encouragingly, 90.9% of newborns had an Apgar score of 7 or higher. While none of the mothers succumbed to the disease, 4.2% experienced major morbidities, and 3.3% required intensive care unit (ICU) interventions [41]. These findings underscore the potential risks associated with SARS-CoV-2 infection during pregnancy, even in largely asymptomatic individuals.

Another study examined mortality rates and reported that maternal and neonatal deaths were 5% and 6%, respectively [42]. Four years later, another study reinforced concerns about the risks associated with SARS-CoV-2 infection during pregnancy, stating: “*SARS-CoV-2 infected pregnant women were significantly associated with an increased risk of adverse pregnancy outcomes, including preterm labor (13.8% vs. 9.5%, p = 0.033) and meconium-stained amniotic fluid (8.9% vs. 5.5%, p = 0.039). The risk of low birth weight (<2500 g) (10.5% vs. 6.5%, p = 0.021) and Apgar score < 8 at 1 min (9.2% vs. 5.8%, p = 0.049) was significantly increased in neonates born to COVID-19 positive mothers compared to their counterparts*” [43]. These findings emphasize a critical point: pregnancy complicated by SARS-CoV-2 infection is a major concern not only for the mother and her family but also for the healthcare professionals responsible for managing the course of treatment.

Recent research [23,30,31,32,33,34,35,36,37,38,39,40,41,42,43] has drawn parallels between SARS-CoV-2 and poliovirus—both RNA viruses historically associated with similar neurological syndromes, as well as respiratory and gastrointestinal complications. The studies suggest that these viruses may first infect bacteria within the human gut microenvironment, classifying them among pathogens that exhibit bacteriophagic behavior. In addition, the researchers observed that in the presence of SARS-CoV-2, bacteria within the human microbiome continuously produce proteins homologous to toxic sequences, better known as toxin-like peptides (P). Notably, this production does not cease immediately following the disappearance of a detectable viral load, raising new concerns about the prolonged effect of infection [44].

These toxin-like peptides (P) were found to be elevated in patients experiencing the acute phase of COVID-19 [31]. A recent study [32] observed that early intervention to reduce viral replication and bacterial toxigenic production during the initial days of COVID-19 [42] resulted in improved survival rates, faster recovery, and consistently high oxygen saturation levels, reducing concerns related to infection management. The management of COVID-19 during pregnancy requires a thorough understanding of the underlying pathophysiological mechanisms that may affect maternal and fetal health. Therefore, the purpose of this paper is to present a comparative analysis of three pregnancies, focusing on the differences in clinical management and outcomes.

The first two cases involve the same unvaccinated mother under similar baseline conditions—one pregnancy occurred before the pandemic (2018–2019) and the second during the pandemic, with a confirmed SARS-CoV-2 infection at 20 weeks gestation (2022–2023). The third case describes a second mother who received two doses of the mRNA vaccine (Pfizer–BioNTech), with the second dose administered during pregnancy. She contracted COVID-19 one month later within the same time period. A comparative analysis of these two women—one unvaccinated and one vaccinated—is presented in the following tables. Additionally, all three pregnancies are evaluated based on ultrasound findings and fetal growth patterns, providing insight into the potential impacts of COVID-19 infection and vaccination during pregnancy.

## 2. Materials and Methods

In paper [32], we published the results of a clinical trial involving 211 cases, investigating the use of amoxicillin and clavulanic acid in combination with rifaximin for the treatment of severe acute SARS-CoV-2 infections in both vaccinated and unvaccinated patients. Approval was granted by the ethics committee (approval number CECN/2115, dated 17 May 2023, Campania North), and the trial was conducted in accordance with the Helsinki Declaration. In the same paper, we also outlined the limitations of antibiotic use and emphasized the need to evaluate a vaccine strategy that is more aligned with the pathogenesis of the bacteriophage-like mechanism. We reiterate here the rationale briefly introduced in that section. Through investigations involving molecular genetics, electron microscopy, immunofluorescence microscopy, and the final technique of using the ^15^N nitrogen radioisotope in bacterial cultures via mass spectrometry analysis, SARS-CoV-2 was initially identified as a viral agent exhibiting bacteriophage-like behavior. To prevent any misinterpretation of our findings, we also re-examined poliovirus derived from fecal samples of patients with poliomyelitis. Through radioisotope assay, we observed that the poliovirus also exhibits the same bacteriophage mechanism. For further details on these experiments, please refer to references [23,24,25,26,27,28,29,30,31,32]. The choice of antibiotic combination was informed by the data obtained via references [23,24,25,26,27,28,29,30,31,32], which showed that not all antibiotics are effective in reducing viral replication. In other words, some antibiotics reduce the number of bacterial cells in which the virus replicates while others do not. This finding was particularly positive for amoxicillin. Another key finding reported in reference [25] was that rifaximin proved more effective in reducing toxins production (P). Given the redefinition of SARS-CoV-2 pathogenesis as both a toxicological event and a bacteriophage-mediated process, the combination of rifaximin with amoxicillin and clavulanic acid emerged as the most suitable therapeutic option for severe and emergency cases, regardless of vaccination status. It is also important to highlight that, according to the established medical literature and clinical guidelines, amoxicillin is considered safe for use during the second and third trimesters of pregnancy, even for minor infections such as dental abscesses. Furthermore, rifaximin stands out among antibiotics for its unique action at the epithelium–microbiota interface and its minimal systemic absorption, as supported by both the scientific literature and pharmaceutical product documentation. The clinical cases discussed in this study are outlined as follows: Case 2.1: Pregnancy of a mother unvaccinated against SARS-CoV-2 before the pandemic; Case 2.1.2: The same mother contracts SARS-CoV-2 at 20 weeks of gestation during the Omicron variant wave; Case 2.2: Pregnancy of a mother vaccinated with the Pfizer–BioNTech vaccine who contracts SARS-CoV-2 at 23 weeks of gestation, one-month post-vaccination.

The mothers came from southern Italy and became ill during the pandemic period. Informed consent from them was obtained according to local legislation. The approval of the ethics committee was registered at CECN/2115, dated May 17, 2023, Campania North, and it was conducted according to the Helsinki Declaration.

### 2.1. Mother Not Vaccinated Against COVID-19

#### 2.1.1. First Case: Pre-Pandemic COVID-19, 2018–2019 (Figure 1)

Caucasian woman, height 167 cm, BMI (mass body index) of 22.23 in 2018, age 35 years, second full-term pregnancy of her life.

Past medical and gynecological history: interatrial septal aneurysm, previous foramen ovale, on cardioaspirin treatment for the duration of each pregnancy. Four pregnancies, including three at term and one miscarriage during the first trimester. Between the second and third pregnancies, she was diagnosed with Monoclonal Gammopathy of Uncertain Significance (MGUS) of IgA light chains.

The first pregnancy (2011–2012) was uncomplicated, and the mother had a BMI of 19.36. The infant was delivered by planned caesarean section at 38 weeks due to maternal cardiovascular risk. Figure 1 shows the ultrasound monitoring throughout the pregnancy, while Table 1 shows the fetal values. The newborn had a birth weight of 3100 g, with Apgar scores of 9 at both 1 min and 5 min.

**Figure 1 biomedicines-13-01858-f001:**
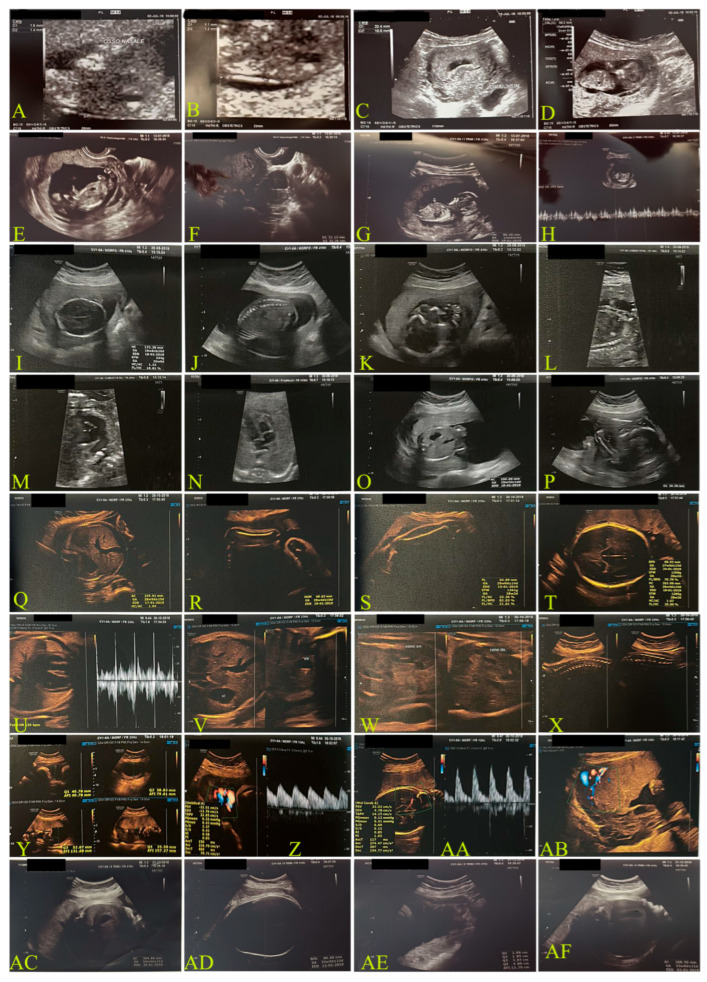
Ultrasound monitoring of case 2.1.1: an unvaccinated 35-year-old mother throughout her 2018 pre-pandemic pregnancy. Letters from (**A**–**U**) describe the various weeks of observations. More details are provided in Table 1, which uses abbreviations. Examinations were performed using a transvaginal probe, 6.5 MHz. (**A**–**D**): estimated amenorrhea (ea.) 11 weeks (w) + 6 days (d); (**E**–**H**): ea. 13 w + 1 d; (**I**–**P**): ea. 20 w + 2 d; (**Q**–**AB**): 28 w; (**AC**–**AF**): 34 w + 3 d.

Similarly, the second pregnancy was carried to 39 weeks and also resulted in a planned cesarean delivery.

#### 2.1.2. Second Case: COVID-19 Pandemic, 2022–2023, Omicron Variant, Same Mother as the Above-Mentioned Case (Figure 2)

This pregnancy took place at the age of 39 with a BMI of 23.31.

In October 2022, the Omicron variant was the most common SARS-CoV-2 mutation in Italy [41]. The pregnant woman contracted the infection in a family environment. At the time of acute illness and infection, she was in her 20th week of pregnancy. Fetal measurements are detailed in the ultrasound image captions. The woman tested positive for SARS-CoV-2 using both the Wizbiotech SARS-CoV-2 Antigen Rapid Test and the RealStar^®^ SARS-CoV-2 RT-PCR Kit 1.0 molecular test.

During the first 36 h, she experienced general malaise, a fever of 39 °C, severe pharyngitis, chest pain, a dry cough, myalgia, arthralgia, rapidly decreasing oxygen saturation (92% SO_2_), and panic attacks. Based on our previous studies, treatment was started immediately with amoxicillin and clavulanic acid twice daily for six days and rifaximin 400 mg daily for three days. In the first 48 h, the patient also experienced an increase in systolic blood pressure to 160 mmHg and in diastolic blood pressure to 95 mmHg, which was promptly controlled with a single 10 mg dose of propranolol.

On day 10, the patient tested negative for SARS-CoV-2, although clinical recovery and physical well-being were achieved by day 5 post-test.

**Figure 2 biomedicines-13-01858-f002:**
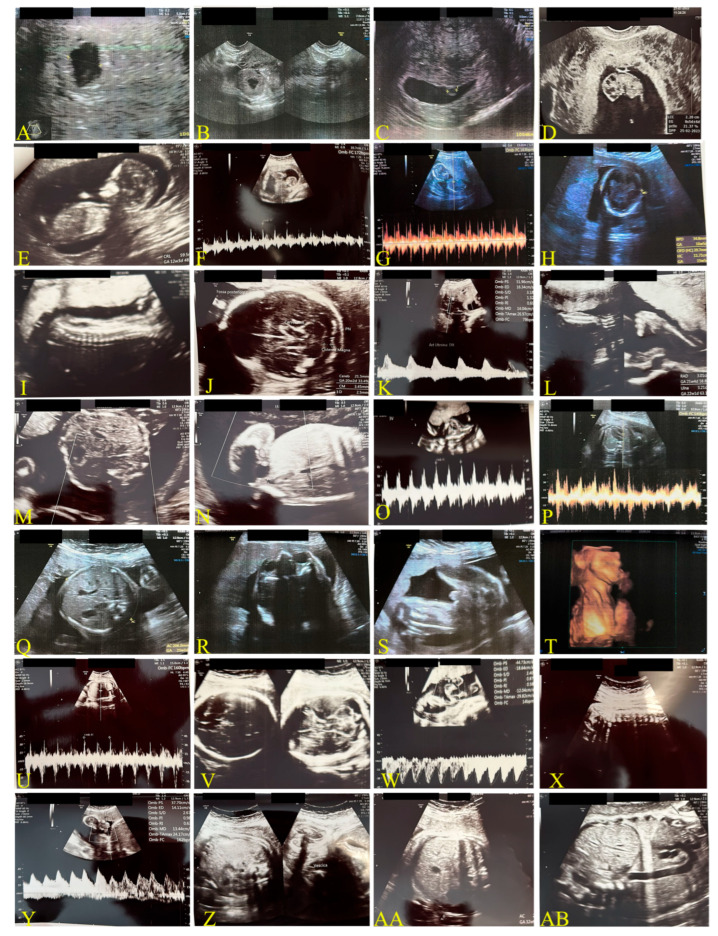
Fetal monitoring for case 2.1.2. Transvaginal images. Unvaccinated mother, 39 years old, transvaginal probe, 6.5 MHz; period 2022–2023, Omicron Variant. (**A**,**B**): newly formed gestational chamber (Date: 24 June 2022); (**C**): gestational chamber after 3 weeks (w) (13 July 2022); (**D**): beginning of visible fetal development (21 July 2022), ea. 9 w + 1 d; (**E**,**F**): ea. 12 w + 2 d; (**G**,**H**): ea. 15 w + 6 d; (**I**–**O**): ea. 21 w + 1 d post-COVID-19 infection; (**P–T**): four-week later images in slide (**I**–**O**); (**U**–**X**): ea. 28 w + 1 d; (**Y**–**AB**): 32 w + 1 d. For further details, refer to the text.

### 2.2. Mother Vaccinated Against COVID-19

#### Third Case: Pandemic Period, December 2021, Delta Variant (Figure 3)

A 36-year-old Caucasian woman, 165 cm in height, with a BMI of 28.65, received two doses of the Pfizer–BioNTech vaccine during early pregnancy. She developed COVID-19 one month after the second dose.

Medical and gynecological history: The patient had had four pregnancies, two of which progressed to full term. She had a medical history of recurrent migraines associated with vertigo syndrome. Notably, during her first full-term pregnancy, she did not experience any migraine or vertigo episodes, despite receiving treatment with cardioaspirin followed by heparin. In her second full-term pregnancy, she continued cardioaspirin therapy for cardiovascular and neurological indications.

At 23 weeks of pregnancy, she tested positive for SARS-CoV-2 via molecular (Viasure real-time PCR detection kit-Cortest Biotec) and antigenic testing (Panbio COVID-19 Ag Rapid test Abbott). The infection occurred four weeks after her second dose of the Pfizer–BioNTech vaccine. Her symptoms were similar to those of the unvaccinated mother, except that she did not receive antibiotic therapy. Instead, she was treated only with anti-inflammatory medications, such as ibuprofen 400 mg daily for a few days (3), in combination with paracetamol 1000 mg daily for 5 days.

She tested negative 20 days after infection, but her symptoms persisted for two weeks. The pregnancy resulted in a preterm birth at 37 weeks by cesarean section. Figure 3 shows ultrasonographic monitoring during pregnancy, while Table 1 reports the ultrasonographic measurements of the case. The following tables compare the ultrasound monitoring and fetal measurements in the three pregnancies.

**Figure 3 biomedicines-13-01858-f003:**
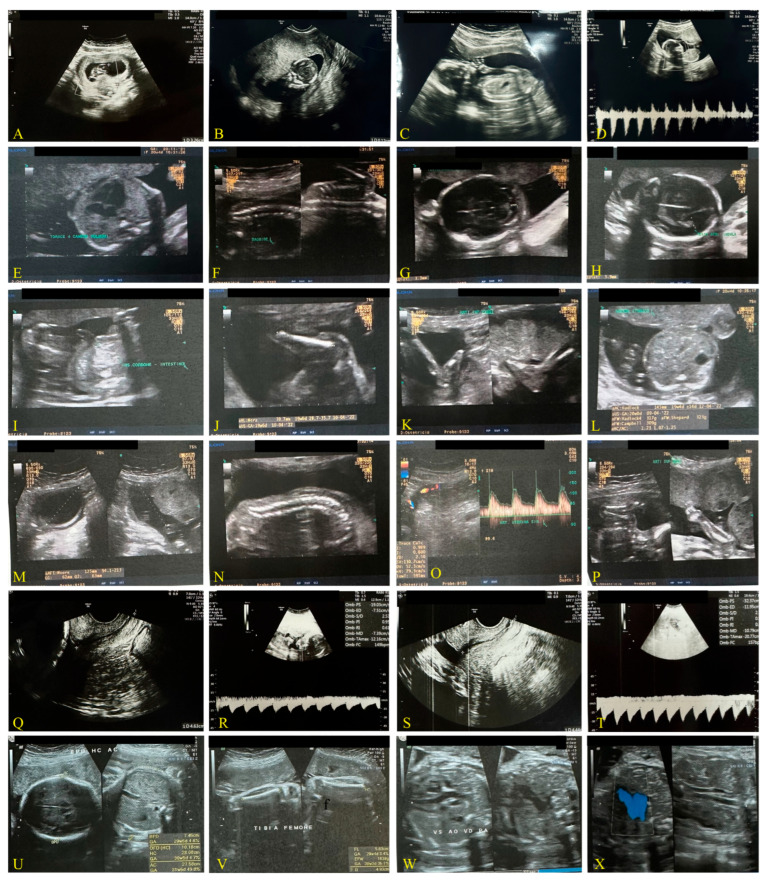
Transvaginal ultrasound images, 6.5 MHz. Vaccinated mother, case 2.2, with two doses of mRNA BioNTech vaccine COVID-19, Period 2021–2022, delta variant. (**A**,**B**): ea. 9 w + 3 d; (**C**,**D**): ea. 16 w + 5 d; (**E**–**P**): ea. 20 w + 5 d; (**Q**,**R**): ea. 23 w; (**S**,**T**): post-COVID-19 infection that occurred 1 month after the second dose of (**U**–**X**): ea. 31 w + 5 d.

### 2.3. COVID-19 Reinfection of the Mother Not Vaccinated During the Breastfeeding

Reinfection (second infection) of the unvaccinated mother (2.1 and 2.2 cases) occurred during breastfeeding, August 2023.

The baby (case 2.1.2.) was born by caesarean delivery in February 2023 at the gynecology department of the Agostino Gemelli IRCCS University Polyclinic Foundation Hospital in Rome, Italy. The mother had undergone two previous cesarean sections. The newborn had a birth weight of 3775 g and an Apgar score of 9 at 1 and 5 min. The infant was exclusively breastfed and had received all required vaccinations according to the Italian national vaccination schedule. In August 2023, the mother experienced a reinfection with COVID-19, confirmed by both rapid antigen and molecular tests. Her symptoms were similar to those of her first infection. She followed the same antibiotic protocol as before, but for a shorter duration of two days, until her symptoms were completely resolved. Throughout the acute phase and recovery, the mother continued breastfeeding, taking precautions such as frequent hand washing and wearing a mask. The child remained asymptomatic and did not contract the infection.

## 3. Results

Table 1 presents the various fetal growth indices assessed in the ultrasonographic evaluation of the three cases. The parameters include heart rate, biparietal diameter (BPD), head circumference (HC), femur length (FL), abdominal circumference (AC), estimated fetal weight, and corresponding weight percentile.

Table 2, on the other hand, shows the values for case 2.1.2: a pregnancy involving an unvaccinated mother who contracted the infection at 20 weeks.

In contrast to Case 2.2, where the mother was vaccinated, fetal growth demonstrated higher values, despite the severe clinical presentation of the disease, as illustrated in Figure 4, Figure 5 and Figure 6.

In the case presented (2.1.2), when COVID-19 infection occurred around 20 weeks, what was observed was a slight and temporary slowing of fetal heartbeats per minute, with the recovery of all growth values in accordance with estimates through treatment of the bacteriophage mechanism (Figure 4). In Figure 5, the figure is best seen by showing how early intervention results in a resumption of growth estimates by disrupting the bacteriophage mechanism underlying the viral infection.

The mothers in Cases 2.1.1 and 2.1.2 shared the same baseline characteristics, as detailed in the Section 2, including BMI and routine pharmacological treatment for pre-existing conditions. There was no significant difference in fetal growth percentiles between the unvaccinated mother’s two pregnancies: one in 2019 (pre-pandemic) and the other in 2022, during which she contracted SARS-CoV-2 while pregnant. However, intervention targeting the unvaccinated bacteriophage mechanism was implemented in the latter case.

In Case 2.1.2, ultrasound evaluation confirmed that the proposed treatment did not cause any significant deviation from normal values in key parameters such as biparietal diameter (BPD), head circumference (HC), femur length (FL), and abdominal circumference (AC). The only parameter showing a slight, temporary reduction was fetal heart rate (BPM), which normalized following intervention addressing the bacteriophage mechanism (Figure 5). Notably, Case 2.1.2 exhibited (Figure 5) a higher overall fetal growth index (Figure 6). In their work, Pistollato et al. highlighted that the association of S (spike protein) and P (toxin-like peptides derived from the gut microbiome) is detrimental to neuronal development in vitro. In the work of Petrillo et al., we evidenced that patients with SARS-CoV-2 infection have elevated P levels. In studies [23,24,25,26,27,28,29,30,31,32], it is shown that the combination of two antibiotics causes a decrease in viral replication (S) and P levels.

Given that Cases 2.1.1 and 2.1.2 involve the same mother under identical baseline conditions, and the only differing factor is the SARS-CoV-2 infection during the 2.1.2 pregnancy, which led to increased levels of viral (S) and toxicological (P) markers, it can be concluded that the combination of the two antibiotics had a beneficial effect in reducing both S and P. Additionally, Case 2.2 shares similar clinical characteristics with Cases 2.1.1 and 2.1.2, including BMI, prior medication use, age, and geographical location. However, following a SARS-CoV-2 diagnosis, treatment with non-steroidal anti-inflammatory drugs (NSAIDs) alone—unlike the in vitro findings reported by Pistollato et al.—did not succeed in suppressing the replication of either viral (S) or toxicological (P) markers. In detail, Figure 6 shows how fetal growth values are preserved in the expected estimates regardless of vaccine status if viral bacteriophage mechanisms are absent (case 2.1.2) or are controlled with the proposed drug therapy (case 2.1.1).

## 4. Discussion

The case report presented is consistent with previous studies [32] and reinforces the findings that there are no substantial differences between a pre-pandemic pregnancy and a pregnancy in a mother with severe COVID-19, provided the infection is managed according to established protocols. This is consistent with previous research emphasizing the bacteriophage mechanism of SARS-CoV-2 [32]. Numerous studies have highlighted the potential negative impact on fetal growth if COVID-19 infection occurs in the second trimester, particularly if not properly managed [36,37,39]. One study reported higher ACE2 receptor expression in pregnant women infected with COVID-19 compared to healthy mothers [47,48]. Some researchers suggest that ACE2 receptor expression plays a role in the release of SARS-CoV-2, based on evidence from in vitro placental models [49,50].

An analysis [51] of differentiated 3D neuronal cells (neurospheres) derived from neuronal stem cells showed the effects of a combination of spike protein (S) and toxin-like peptides (P) over a 72-h period. Both 2-week-old and 8-week-old differentiated neurospheres were exposed to the mixture. The data revealed, as reported by the authors, the following:

“*Spike protein and toxin-like peptides at non-cytotoxic concentrations differentially perturb the expression of SPHK1 (Sphingosine kinase 1), ELN (Elastin), GASK1B (Golgi-associated kinase 1B), HEY1 (Hes Related Family BHLH Transcription Factor With YRPW Motif 1), UTS2 (Urotensin-II), ACE2 (Angiotensin-Converting Enzyme 2), and several neuronal, glial, and neural stem cell (NSC)-related genes critical during brain development*.”

This study specifically examined the combined effects of spike protein (S) (Sigma-Merck, cat SAB5700592) and toxin-like peptides (P) derived from fecal bacterial cultures from SARS-CoV-2-infected individuals and demonstrated a synergistic impact on neuronal function. Additionally, the combination of spike protein (S) and toxin-like peptides (P) significantly reduced spontaneous electrical activity in long-term differentiated neuronal cultures. While 0.5 μg/mL of P alone did not alter spontaneous electrical activity, its combination with S resulted in a more than 50% decrease in neuronal electrical activity [51]. Viral infections during pregnancy have long been a concern, and many studies have investigated the mechanisms of maternal–fetal transmission and the potential consequences for fetal development [52,53]. Previous research has linked infections during pregnancy to an increased risk of preterm birth and fetal malformations. However, bacterial co-infections are of even greater concern.

Evidence suggests that bacterial over-infection, resulting from the depletion of beneficial gut microbiota, is common during COVID-19 infection. The growing recognition of the bacteriophagic nature of SARS-CoV-2, or other viruses, highlights the need for further investigation and increased awareness within the scientific community [27,30]. Several studies have established that proteins, like immunoglobulins, antibodies, and smaller proteins, can be transferred from the mother to the fetus through the placenta, depending on their molecular size. This underscores the importance of toxin-like peptides, not only for their deleterious effects on neuronal and muscle electrical conduction but also for their role in activating specific genes that compromise neuronal viability in newborns [51].

Fetal growth monitoring through ultrasound diagnostics offers an indirect yet valuable assessment of fetal health. Ultrasonography represents a significant scientific advancement in pregnancy management, and its routine use during viral infections enhances clinical decision-making by enabling timely interventions.

As demonstrated in this case report, the child born to a mother infected with COVID-19, who exhibited a high growth percentile (Figure 4, Figure 5 and Figure 6), illustrates how closely spaced ultrasound examinations can effectively track the impact of post-infection therapy. Key fetal growth parameters—including estimated weight, head circumference, abdominal circumference, and femur length—showed a temporary slowdown due to the infection, as depicted in Figure 4 and Figure 5. However, these growth patterns promptly recovered following therapeutic intervention. Conversely, the case of the mother who was vaccinated during pregnancy and became infected one month later highlights a different outcome. Despite alternative management approaches, the pregnancy resulted in preterm birth, consistent with findings from other studies [39,40], and a lower fetal growth percentile. This suggests that not only the spike protein generated through bacteriophage mechanisms [27,30] and the related toxin-like peptides but also the additional spike protein introduced by vaccination may contribute to the overall effect. This points to a possible additive impact, particularly given that the spike protein produced by mRNA vaccines is more stable and resistant due to specific amino acid modifications designed to maintain its prefusion conformation, as demonstrated in a study examining its half-life compared to the wild-type protein, as reported [54] by the comment on our recent study finding that the spike protein transcript derived from the mRNA vaccine (with the double proline, PP, modification) persisted in half of the subjects for up to 180 days post-vaccination. Unfortunately, this type of vaccination does not allow precise quantification of the mRNA-induced protein dosage, resulting in significant uncertainty compared to traditional vaccines that use only the protein antigen for immunization.

The second COVID-19 infection in the unvaccinated mother did not result in infection in the child, probably because of the passive immunity transferred through breastfeeding. A recent study [55] of 187 COVID-19-positive women who donated blood and breast milk during the acute phase of infection found no traces of the virus in breast milk. However, researchers identified immunoglobulins A, G, and M in both colostrum and mature milk, with immunoglobulin A being particularly abundant in colostrum. These findings highlight the protective role of immunoglobulins in breast milk. In the present case, despite the second maternal infection, the child—who was six months old and still breastfeeding—remained uninfected, further supporting the immunoprotective properties of maternal milk [55].

## 5. Conclusions

Maternal distress often translates into fetal distress, making timely and appropriate case management critical to achieving favorable outcomes.

Numerous studies have documented increased rates of preterm birth and pregnancy complications associated with SARS-CoV-2 infection. In light of the findings outlined in our previous work and reinforced by this case series, there is an urgent need for a vaccine strategy that takes into account the bacteriophagic nature of SARS-CoV-2. Viruses that exhibit both prokaryotic and eukaryotic replication behaviors present unique challenges for vaccine and treatment development [56]. In particular, bacterial toxins continuously produced following viral infection can be effectively controlled through the use of an appropriate antibiotic combination. The microbiome is also a crucial factor in the management of infections during pregnancy. It is important to emphasize that a deeper understanding of these mechanisms could offer a more accurate explanation for an alternative approach to vaccination—one involving attenuated virions, as was successfully implemented in the past for polio. This approach did not rely solely on achieving high antibody levels as the endpoint. If bacteria within the microbiota are involved, then vaccination strategies should first address these bacteria or enhance resistance to them; minimizing the production of toxin-like peptides during bacterial infection of the microbiome is a critical issue that should not be overlooked, particularly in pregnant women. This underscores the importance of careful monitoring and timely intervention in cases of COVID-19 infection during pregnancy. Combined with the use of ultrasonography throughout gestation, this knowledge enables a more comprehensive assessment of neonatal indices and supports informed clinical decisions to benefit both mothers and their fetuses.

This study highlights the importance of understanding the bacteriophage mechanisms of SARS-CoV-2 and other potential viruses. Such knowledge equips clinicians with a broader perspective to evaluate and manage maternal and fetal health with appropriate care during acute infections.

Finally, we emphasize the importance of protecting neuronal and fetal development. By avoiding an increased bacterial and viral protein load during pregnancy and monitoring gestational ultrasound parameters, it is possible to monitor the outcomes of an infection that affects both mother and fetus.

## Figures and Tables

**Figure 4 biomedicines-13-01858-f004:**
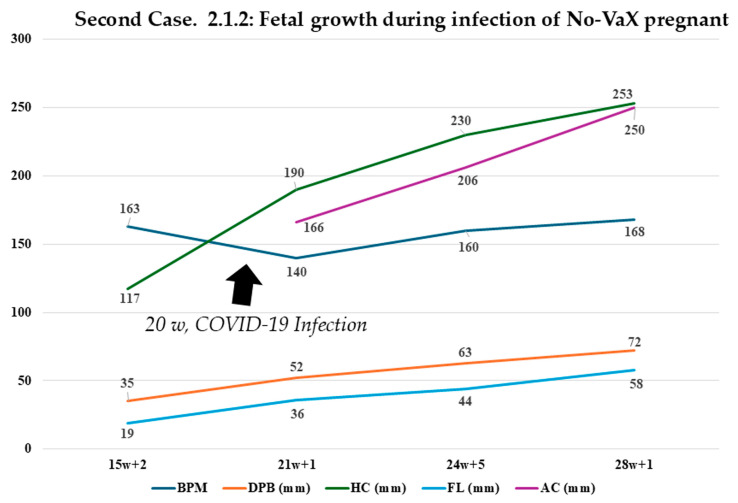
Monitoring of fetal growth in case 2.1.2. Representation of fetal growth before and after the COVID-19 infection in case 2.1.2., involving an unvaccinated mother. The ultrasound study shows that, with the proposed treatment, there was no significant deviation from the normal range for values such as DPB, HC, FL, and AC. The only value that showed a slight and temporary slowdown was the BPM (beats per minute).

**Figure 5 biomedicines-13-01858-f005:**
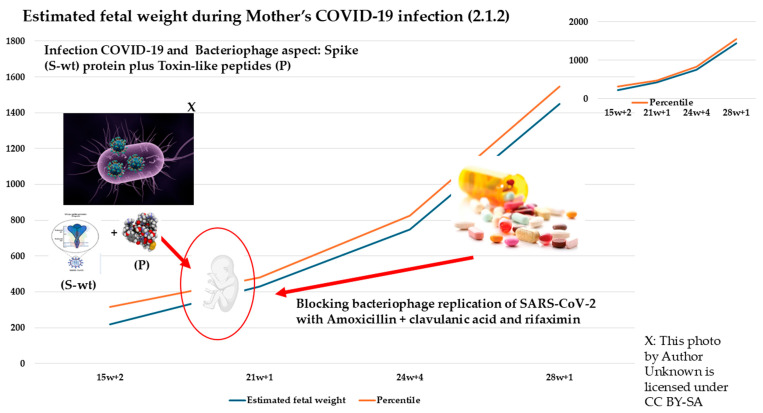
The image shows the risk obtained for the additive effect of the viral spike protein (S), the spike protein (PP) induced by the vaccination and the toxin-like peptides (P) produced as a result of the bacteriophage mechanism and present in the circulation. The graph showing the estimated fetal weight of the case 2.1.2 stimated before and after treatment. The data show that with the proposed treatment, which blocks viral replication (S) on bacterial cells and blocks the production of toxin-like peptides (P) by bacteria of gut microbiome, it was possible to bring the estimated weight values back to the expected standards, as monitored by ultrasound.

**Figure 6 biomedicines-13-01858-f006:**
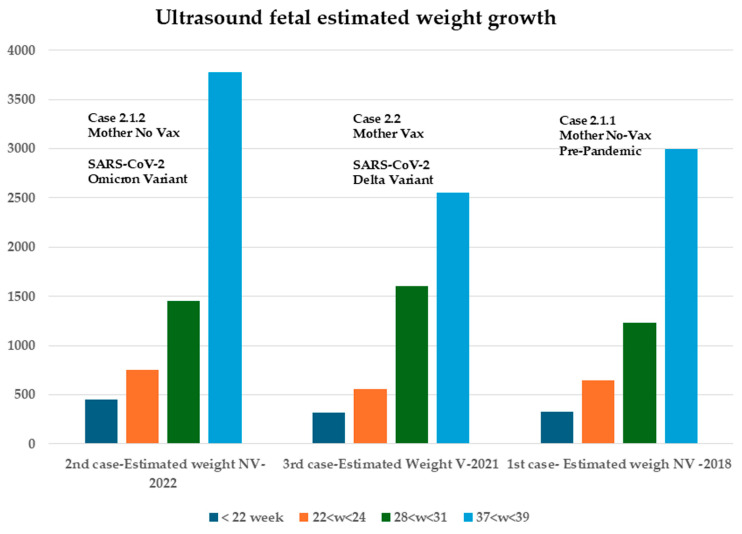
Fetal ultrasound of estimated growth in weight. Difference in ultrasound values of the 3 pregnancies of the fetuses. On Y-axis, there is the weight expressed in grams, and on X-axis the weeks of pregnancy. Case 2.1.2. shows a higher growth index.

**Table 1 biomedicines-13-01858-t001:** Description of the differences in values in ultrasound images of the three pregnancies. Two mothers were infected during the pandemic [45].

Period	2018–2019 Mother UnVax	2022–2023 Mother UnVax Omicron Variant	2021–2022 Mother Vax Delta Variant
Vaccination mRNA	No	No	2 doses—second during pregnancy, 1 month before infection
Newborn gender	Female	Male	Female
Birth weight	3100 g (32.3° pc)	3775 g (79.1° pc)	2550 g (22.8° pc)
Birth length	50 cm	53 cm	46 cm
Cranial circumference	32 cm	35 cm	33.5 cm
Apgar first-minute index	9	9	**8**
Apgar index at 5 min	9	9	9
Surgery	Cesarean section under general anesthesia	Cesarean section under general anesthesia	Cesarean section under locoregional anesthesia
Mother’s weight at term	78 kg	78 kg	78 kg
Mother’s BMI at term pregnancy	27.97	27.97	31.6
Infections during pregnancy	None	COVID-19	COVID-19
Therapy for infections	-	Amoxicillin plus clavulanic acid + rifaximin	Paracetamol plus ibuprofen
Comorbidity 1 Comorbidity 2	Pervious foramen ovale; interatrial septal aneurysm	Pervious Foramen ovale; interatrial septal aneurysm	Migraine
Comorbidity 3		MGUS	
Standard therapy	Cardioaspirin	Cardioaspirin	Cardioaspirin
Pregnancy weeks	39 weeks	39 weeks	37 weeks + 4 days
**Weeks of gestation and Ultrasound data**	**20 + 2 days**	**21 + 1 days**	**20 + 4 days**
Biparietal diameter (BPD)	47 mm	52 mm	46.4 mm
Occipital-frontal diameter (OFD)	61 mm	67 mm	-
Head circumference (HC)	-	190 mm	178 mm
Thermal index for bone (TIB)	-	31 mm	-
Fetal corpus callosum (CC)	172 mm	190 mm	-
Cerebellum transverse diameter	20.5 mm	21.5 mm	-
Lateral trigon diameter (Lateral Ventriculum)	6.8 mm	6 mm	-
Abdominal circumference (AC)	150 mm	166 mm	145 mm
Femur length (FL)	32 mm	36 mm	32.3
Occiput lateral (head down, facing your side) (OL)	-	33 mm	-
Heart rate	151 bpm	140 bpm	140 bpm
Estimated weight	330 g	430 g	320 g
**Weeks of gestation and US data**	**29 days**	**28 + 1 days**	**31 days**
Biparietal diameter (BPD)	69 mm	72 mm	79 mm
CC	258 mm	253 mm	280 mm
Cerebellum transverse diameter	30 mm	21.5 mm	-
AC	246 mm	250 mm	273 mm
LF	49 mm	58 mm	56 mm
Estimated weight	1230 g	1451 g	1600 g

**Table 2 biomedicines-13-01858-t002:** Estimated growth during the period of COVID-19 infection of Second Case [46].

Week	15 + 2 Days	21 + 1 Day	24 + 5 Days	28 + 1 Day
MAF	YES	YES	YES	YES
Heart rate	163 bpm	140 bpm	160 bpm	168 bpm
Biparietal Diameter (DPB)	35 mm (68° pc)	52 mm	63 mm	72 mm
Head circumference (HC)	117 mm (31° pc)	190 mm	230 mm	253 mm
Femur length (FL)	19 mm (45° pc)	36 mm	44 mm	58 mm
Abdominal circumference (AC)	-	166 mm	206 mm	250 mm
Estimated fetal weight	218 g	430 g	750 g	1451 g
Percentile of weight	97.5th	67.3th	52.5th	97.5th

## Data Availability

All data are reported in the present manuscript.

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
