# Peer review of "New Neonatal and Prenatal Approach to Home Therapy with Amoxicillin, Rifaximin, and Anti-Inflammatory Drugs for Pregnant Women with COVID-19 Infections—Monitoring of Fetal Growth as a Prognostic Factor: A Triple Case Series (N.A.T.H.A.N.)"

_biomedicines, 2025, doi:10.3390/biomedicines13081858_

Round 1
Reviewer 1 Report
Comments and Suggestions for Authors
Review Report Form
Manuscript ID: biomedicines-3699451
Title: New neonatal and prenatal approach to home therapy with amoxicillin, rifaximin, and anti-inflammatory drugs for pregnant women with COVID-19 infections – monitoring of fetal growth as a prognostic factor: a triple case series (N.A.T.H.A.N.)
- Overall Recommendation
Major Revision
- Comments to the Author
General Overview:
This manuscript describes a triple case series evaluating pregnancy outcomes in the context of COVID-19 infection, with a focus on fetal ultrasound parameters and proposed benefits of early antibiotic therapy.
Major Concerns:
- Non-compliance with MDPI Formatting and Structure
- The manuscript does not follow the standard MDPI template.
- The abstract is unstructured; please use Background, Methods, Results, Conclusions
- Figures, tables, and sections should follow the MDPI layout guidelines.
- References Not in Required MDPI Style
- In-text citations are incorrectly placed (e.g., “[40.”).
- The reference list does not follow Vancouver/MDPI format. Please reformat all citations accordingly.
- Scientific Overstatements and Speculative Content
- The paper repeatedly suggests that SARS-CoV-2 has bacteriophage-like properties and that this justifies antibiotic therapy. These claims are speculative and not supported by current consensus.
- The conclusion of treatment efficacy is overstated for a case series of three pregnancies.
- Unbalanced Vaccine Discussion
- The paper implies a negative effect of mRNA vaccination based on a single case, without considering large cohort evidence.
- A balanced and data-driven discussion is required, citing broader literature.
Minor Issues:
- Numerous grammatical and syntax errors. A full language revision is necessary.
- Figure legends lack sufficient detail (gestational age, measurement units, percentile references).
- Inconsistent use of abbreviations (e.g., BPD, FL, AC) should be corrected.
Summary of Required Actions:
- Reformat the manuscript using the official MDPI template.
- Rewrite the abstract using the required structured format.
- Revise all references and in-text citations to match MDPI/Vancouver style.
- Remove or reframe speculative mechanisms (bacteriophage behavior, toxin-like peptides).
- Reword conclusions to reflect observational nature.
- Provide a balanced view of COVID-19 vaccination outcomes in pregnancy.
- Address language and stylistic inconsistencies.
- Comments on the Quality of English Language
Needs major editing
The manuscript contains numerous grammatical errors, run-on sentences, and awkward phrasing that impact clarity and scientific tone. A professional English editing service is recommended.
Reviewer 2 Report
Comments and Suggestions for Authors
Comment 1: There is insufficient evidence to confirm that the pathogenesis of these cases is due to a bacteriophage mechanism. Further basic experiments and ethically approved clinical research are needed to verify this. Additionally, the potential use of amoxicillin and rifaximin in treating COVID-19 in pregnant women also requires further study to confirm the effectiveness and safety.
Comment 2: It is recommended that the original content of the reference article not be quoted directly in lines 130-135, 140-143, and 149-152.
Comment 3: The introduction should be more concise.
Comment 4: Line 192 states that these cases have similar baseline characteristics, but the specific data is not presented in the results.
Comment 5: The title of Figure 1 should be more specific. The content displayed in the figure should be described in both the manuscript and the figure legend.
Comment 6: The diagnostic and therapeutic process of Case 2 should be more specifically described. For example, the laboratory tests, imaging, and therapeutic process of Case 2 should be presented.
Comment 7: Figure 2 should be revised to have a more specific title and figure legend.
Comment 8: I suggest describing the maternal and fetal information in Table 1 separately and highlighting the abnormal results.
Comment 9: The contents of Tables 1 and 2 are not described in detail in the manuscript.
Comment 10: There is insufficient direct data to confirm whether targeting viral replication (S) and bacterial cellular toxin-like peptide (P) production can restore estimated weight values to the expected standard.
Comment 11: Figures 4, 5, and 6 lack titles. The manuscript does not provide detailed information about its related content.
Comment 12: The Conclusion should be shorter.
Round 2
Reviewer 1 Report
Comments and Suggestions for Authors
Please use the Biomedicines template!
Author Response
We thank the reviewer for his/her suggestions and we assure that we have used the Biomedicine template.
Reviewer 2 Report
Comments and Suggestions for Authors
I carefully read lines 447–466 of the latest manuscript and found that they are located in the discussion section. If possible, the descriptions of Figures 4, 5, and 6 should be added in the Results section.
Author Response
We thank the reviewer for his/her suggestions and we have introduced his new advise in lines 425-430 and 458-460.
Many thanks.